# Early Enteral Nutrition (within 48 h) for Patients with Sepsis or Septic Shock: A Systematic Review and Meta-Analysis

**DOI:** 10.3390/nu16111560

**Published:** 2024-05-22

**Authors:** Carlos F. Grillo-Ardila, Diego Tibavizco-Palacios, Luis C. Triana, Saúl J. Rugeles, María T. Vallejo-Ortega, Carlos H. Calderón-Franco, Juan J. Ramírez-Mosquera

**Affiliations:** 1Department of Internal Medicine, School of Medicine, Pontificia Universidad Javeriana, Bogotá 110231, Colombia; datibavizco@husi.org.co (D.T.-P.); lctriana.med@javeriana.edu.co (L.C.T.); 2Department of Obstetrics & Gynecology, School of Medicine, Universidad Nacional de Colombia, Bogotá 111321, Colombia; 3Intensive Care Unit, Hospital Universitario San Ignacio, Bogotá 110231, Colombia; 4Department of Surgery, School of Medicine, Pontificia Universidad Javeriana, Bogotá 110231, Colombia; sjrugeles@husi.org.co; 5Clinical Research Institute, Universidad Nacional de Colombia, Bogotá 11001, Colombia; mtvallejoo@unal.edu.co; 6Department of Internal Medicine, School of Medicine, Universidad del Bosque, Bogotá 11001, Colombia; cacalderon190@gmail.com; 7School of Medicine, Pontificia Universidad Javeriana, Bogotá 110231, Colombia; juanj-ramirez@javeriana.edu.co

**Keywords:** sepsis, enteral nutrition, critical care, systematic review

## Abstract

OBJECTIVE: Medical nutrition therapy provides the opportunity to compensate for muscle wasting and immune response activation during stress and trauma. The objective of this systematic review is to assess the safety and effectiveness of early enteral nutrition (EEN) in adults with sepsis or septic shock. METHODS: The MEDLINE, Embase, CENTRAL, CINAHL, ClinicalTrials.gov, and ICTRP tools were searched from inception until July 2023. Conference proceedings, the reference lists of included studies, and expert content were queried to identify additional publications. Two review authors completed the study selection, data extraction, and risk of bias assessment; disagreements were resolved through discussion. Inclusion criteria were randomized controlled trials (RCTs) and non-randomized studies (NRSs) comparing the administration of EEN with no or delayed enteral nutrition (DEE) in adult populations with sepsis or septic shock. RESULTS: Five RCTs (*n* = 442 participants) and ten NRSs (*n* = 3724 participants) were included. Low-certainty evidence from RCTs and NRSs suggests that patients receiving EEN could require fewer days of mechanical ventilation (MD −2.65; 95% CI, −4.44–0.86; and MD −2.94; 95% CI, −3.64–−2.23, respectively) and may show lower SOFA scores during follow-up (MD −1.64 points; 95% CI, −2.60–−0.68; and MD −1.08 points; 95% CI, −1.90–−0.26, respectively), albeit with an increased frequency of diarrhea episodes (OR 2.23, 95% CI 1.115–4.34). Even though the patients with EEN show a lower in-hospital mortality rate both in RCTs (OR 0.69; 95% CI, 0.39–1.23) and NRSs (OR 0.89; 95% CI, 0.69–1.13), this difference does not achieve statistical significance. There were no apparent differences for other outcomes. CONCLUSIONS: Low-quality evidence suggests that EEN may be a safe and effective intervention for the management of critically ill patients with sepsis or septic shock.

## 1. Introduction

Sepsis is a life-threatening organ dysfunction caused by a dysregulated host response to infection [1]. It is a condition that, when accompanied by persisting hypotension that requires the use of vasopressors and a serum lactate level despite adequate volume resuscitation (e.g., septic shock), has a far bleaker prognosis, with a mortality that exceeds 40% [2]. In 2017, sepsis and septic shock affected 49 million individuals and was related to 11 million potentially avoidable deaths worldwide, accounting for 20% of all annual deaths globally, with an estimated cost of more than USD 32,000 per affected patient [3].

Crystalloids, antibiotics, vasopressors, mechanical ventilation, and prompt admission to the intensive care unit are recognized interventions that improve survival in patients with sepsis and septic shock [2,4]. Besides these, medical nutrition therapy represents another cornerstone that may impact prognosis, favoring rehabilitation and functional status recovery [5,6]. Medical nutrition therapy provides the opportunity to compensate and correct significant muscle wasting [6], oxidative stress, and immune response activation in sepsis or septic shock, counterbalancing the energy deficit [7]. Enteric metabolic support preserves gut integrity and intestinal permeability and contributes to inflammatory response and insulin resistance down-modulation [8].

Although the start of early enteral nutrition (EEN)—defined as being initiated within the first 48 h—has been shown to decrease mortality and improve other relevant clinical outcomes in patients with multiple trauma, traumatic brain injury [9], severe COVID-19 infection [10], severe burn injuries [11], and severe acute pancreatitis [12], there is uncertainty regarding the role of this intervention in patients with sepsis or septic shock [13]. Theoretically, impaired splanchnic perfusion related to sepsis, and especially in septic shock, can potentially represent an extra workload leading to bowel ischemia or necrosis; this, added to the fact that during the first hours after ICU admission, patients experience intense stress-induced endogenous production of the metabolic substrate, may give rise to a theoretical increase in the risk of complications [13].

The current clinical evidence is controversial. One study suggested that increased amounts of calories and protein per day were associated with a decrease in mortality but a potential negative impact on ventilation-free days in septic patients admitted to the intensive care unit [14]. In contrast, another study in which early enteral nutrition was initiated and the energy target was reached within 48 h after admission did not show a difference in survival [15]; a third study [16] showed that the use of early enteral nutrition in patients with septic shock increased mortality and morbidity. Consequently, in view of the knowledge gaps, the existing controversy, and the absence of systematic reviews [13], the objective of this systematic review is to assess the safety and effectiveness of EEN in adults with sepsis or septic shock, regardless of the etiology.

## 2. Materials and Methods

The protocol of this review was designed following the Cochrane Handbook recommendations [17] and the methodological guidance of the Cochrane Methods Group for Non-Randomized Studies for Interventions [18,19] and was registered in advance in Prospero on 7 July 2023 (CRD42023439265), following the PRISMA statement [20]. Because this study is a systematic review, no ethical approval was required. An electronic search was conducted in the MEDLINE, Embase, and CENTRAL databases. CINAHL, ClinicalTrials.gov, and the International Clinical Trials Registry Platform (ICTRP) were searched for ongoing studies through the CENTRAL platform. Gray literature identification was performed from conference proceeding abstracts of the events organized by the Society of Critical Care Medicine (SCCM) and the European Society of Intensive Care Medicine (ESICM), and the reference lists of all included studies and expert content were queried to identify additional relevant publications. There were no language or date restrictions. Databases were searched from inception until 1 July 2023. Details can be found in the supplemental digital content (Appendix A).

Primary outcomes were mortality (e.g., in-hospital, and at 28 and 90 days), days of mechanical ventilation, renal replacement therapy, and feeding intolerance (e.g., diarrhea, ileus, etc.). Secondary outcomes were SOFA score, infectious complications (e.g., ventilator-associated pneumonia), functional status after hospital discharge, and costs derived from the intervention. Inclusion criteria were randomized controlled trials (RCTs) and non-randomized studies (NRSs) comparing the administration of early enteral nutrition (within 48 h) with no nutrition or delayed enteral nutrition (after 48 h) for the management of patients over 16 years of age who had been admitted to the intensive care unit (ICU) for at least 48 h due to sepsis or septic shock, regardless of the etiology.

Two review authors evaluated the limitations of each study independently, and disagreements were resolved through discussion, with a third author being engaged if otherwise needed. For RCTs, judgments were performed using the domains established in the *Cochrane Handbook for Systematic Reviews of Interventions* [17]. For NRSs, the risk of bias was assessed using the ROBINS-I tool [21], which considers flaws attributable to the measurement of outcomes, selection of the reported result, confounders, missing data, departures from intended interventions, selection of participants, and classification of interventions. The available options for risk-of-bias assessments encompassed low, moderate, serious, or critical categories [21].

Search results were arranged using Mendeley software v1.19.8 (Mendeley Ltd., Kidlington, UK), removing duplicated records. All titles, abstracts, full-text assessment, and data extraction of relevant studies were performed by three review authors independently. Disagreements were resolved through discussion. The extracted data contained study characteristics, methodology, setting, participant descriptions (e.g., age, gender, body mass index [BMI], malnutrition prevalence, comorbidities, use of vasopressor or inotropic drugs, severity of illness, sepsis source); details of the intervention (e.g., clinical condition at intervention administration, intake goal defined as kcal/kg/day, and protein) and comparison; and outcomes.

Data were entered into RevMan version 5.4 (The Nordic Cochrane Centre, The Cochrane Collaboration, Copenhagen, Denmark) by two authors and verified for accuracy. For included studies that reported median, minimum and maximum values or first and third quartiles, sample means and standard deviations were calculated according to the method proposed by Wan et al. [22], and when information was not clear, the authors of the original reports were contacted for further information. The results are presented as a summary odds ratio (OR) with 95% confidence intervals (CIs) for dichotomous data and as a mean difference for continuous data. Statistical heterogeneity was evaluated using τ^2^, *I*^2^, and χ^2^ test values. Heterogeneity was considered substantial if the statistical value of *I*^2^ was higher than 40%, if the value of τ^2^ was greater than 0, or if there was a low *p*-value (less than 0.10) in the χ^2^ test.

To avoid unit-of-analysis error in the case of RCTs with multiple, correlated comparisons, the approach suggested by the *Cochrane Handbook for Systematic Reviews of Interventions* was used [17]. A fixed-effect meta-analysis was implemented for combining data when it was reasonable to assume that studies were estimating the same underlying treatment effect. If there was remarkable clinical heterogeneity or substantial statistical heterogeneity, sufficient for expecting that the underlying treatment effects differed between studies, a random-effects meta-analysis would be used to produce an overall summary [17]. For meta-analyses with more than 10 studies, the publication bias was explored by a visual inspection of funnel plot asymmetry. GRADEpro was used to create “Summary of findings” tables (SoF) [23]. The GRADE [24,25] methodology evaluates the overall quality of the body of evidence for each outcome according to risk bias, inconsistency, imprecision, indirectness, and publication bias criteria.

## 3. Results

The search retrieved 3759 references, and 3086 studies were screened after duplicate removal; a total of 37 references were reviewed in full text (Figure 1), 15 studies fulfilled the inclusion criteria [26,27,28,29,30,31,32,33,34,35,36,37,38,39,40], and 22 studies were excluded because they engaged with a different population, administered a distinct intervention or comparison, or implemented a non-relevant design. The details of included and excluded studies can be found in the supplemental digital content (Appendix A).

Five RCTs were retrieved [26,27,28,29,39], and ten studies were NRSs [30,31,32,33,34,35,36,37,38,40], providing a total sample size of 4166 participants. The studies were carried out in Argentina [36], Australia [36], Brazil [36], Canada [36], China [30,32,34,39,40], Greece [36], Hong Kong [36], Japan [33], India [27,28], Mexico [35,36], Malesia [36], Panama [36], Saudi Arabia [36], the United States [29,31,37,38], and the United Kingdom [26,36]. Most of them were single-center studies [26,28,30,31,33,34,37,38,39,40] and recruited their participants in teaching hospitals [27,28,29,33,34,37,38]. All studies were published in English, except for one that was published in Spanish [35]. Studies were sponsored by non-governmental organizations [30,32,34,37,39], academic institutions [29], or by the pharmaceutical industry [26]. Three studies did not receive any funding [33,35,36], and the supporting source was not mentioned in five [27,28,31,38,40].

The included studies recruited patients aged between 18 and 94 years, regardless of gender (20 to 53% women) and body mass index (18 to 40 kg/m^2^). Five studies [31,34,35,36,40] reported arterial hypertension, diabetes mellitus, COPD, cancer, heart failure, and chronic kidney disease as the most prevalent comorbidities among participants, and two studies [29,36] specifically informed the frequency of severe malnutrition in their subjects (20 to 28%). SOFA scores ranged between 6 and 15 points, while APACHE II scores were between 13 and 29 points. The focus of sepsis was almost always abdominal, pulmonary, or urinary; all studies informed the use of vasopressor or inotropic support, but three of them [29,33,37] reported the combined use of two or more supports at the time of administering the intervention (13 to 40%). Five studies preferentially enrolled patients on ventilatory support (72 to 100%) [26,29,32,34,40].

EEN was administered by nasogastric or nasojejunal tubes during the first 24 to 48 h in all studies, allowing for the initiation of this support when inotropic and vasopressor medications were at stable, low, or decreasing doses. Two studies [29,37] specifically mentioned the initiation of enteral nutrition at trophic doses upon admission to the ICU, with an increase in infusion rate as vasopressor support was withdrawn [29]. The calorie goal ranged between 20 and 25 kcal/kg/day, except for four studies that administered a hypocaloric [33,35,36] or hypercaloric diet [27]. Protein concentrations fluctuated between 1.2 and 2.0 g/kg/day, except in two studies that administered a low-protein diet [35,36]. One study added glutamine, arginine, glycine, EPA, and DHA to the intervention group [26]. Two studies allowed for the initiation of complementary parenteral nutrition after four days if necessary [30,32]. All studies were characterized by the start of EEN at a slow infusion rate, with a gradual increase according to tolerance. In five studies, information on nutritional support was limited [28,31,36,38,40]. Control groups received nutritional support after 48 h, and three studies allowed intravenous administration of dextrose-containing fluids during this period [26,27,28].

Two RCTs were appraised as low risk of bias in relation to the randomization process [29,39], while there were “some concerns” for three studies in this regard given the lack of clear information about the method implemented for random sequence generation and allocation concealment [26,27,28]. All RCTs were assessed as “some concerns” for the domain bias due to deviations from intended interventions and bias in the measurement of the outcome, given that participants, trial personnel, and outcome assessors were aware of the patients assigned [26,27,28,29,39]. Included RCTs were graded as low risk of bias for missing outcome data and the selection of the reported results domains, as outcome data were available for nearly all participants, and all reported results corresponded to all intended outcome measurements [26,27,28,29,39].

All NRSs were judged to be at high risk for the confounding domain [9,30,31,32,33,34,35,36,37,38,40], while all except one [37] of the studies were appraised as low risk of bias for participant selection. Five studies [31,33,35,38,40] were graded as “no information” for classification of intervention criteria, and all studies were assessed as low risk of bias for deviations from the intended interventions, missing data, and measurement of outcomes domains. Only one study registered the protocol [36], making the selection of the reported results unclear for the rest of the included studies. The quality of evidence table, along with the respective “SoF” table, can be found in the supplemental digital content (Appendix A).

### 3.1. In-Hospital, 28-, and 90-Day Mortality

Five RCTs [26,27,28,29,39] and ten NRSs [31,32,33,34,35,36,37,38,39] reported in-hospital and 28-day mortality. Even though a lower in-hospital mortality rate is shown in patients with EEN in both RCTs (OR 0.69; 95% CI, 0.39–1.23) as well as NRSs (OR 0.89; 95% CI, 0.69–1.13) (Figure 2), this difference does not achieve statistical significance. There was no apparent difference in mortality at 28 days from RCTs (OR 1.06; 95% CI, 0.45–2.46) or NRSs (OR 0.89; 95% CI, 0.74–1.08) (Appendix A). The quality of evidence was graded as low. None of the studies reported 90-day mortality.

### 3.2. Mechanical Ventilation, Renal Replacement Therapy, and SOFA Score

Three RCTs [26,29,39] and eight NRSs [30,31,33,34,35,36] analyzed these outcomes. Low-confidence evidence from the RCTs and NRSs suggests that patients who receive early enteral support could require fewer days of mechanical ventilation (MD −2.65; 95% CL, −4.44–0.86; and MD −2.94; 95% CI, −3.64–−2.23, respectively) (Figure 3) and may show lower SOFA scores during follow-up (MD −1.64 points; 95% CI, −2.60–−0.68; and MD −1.08 points; 95% CI, −1.90–−0.26, respectively) (Figure 4) without an apparent difference in the requirement of renal replacement therapy (OR 0.92; 95% CI, 0.41–2.03; and OR 0.89; 95% CI, 0.46–1.73, respectively) (Figure 5).

### 3.3. Adverse Events of Enteral Nutrition

Three RCTs [26,28,37] and one NRS [37] reported the incidence of ventilator-associated pneumonia, diarrhea, and ileus. Low-quality evidence from the RCTs and NRS suggests that the administration of EEN could increase the incidence of diarrhea episodes (OR 2.23, 95% CI 1.15–4.34) (Figure 6) without an apparent difference in the proportion of patients with ileus (OR 0.57, 95% CI 0.06–5.04) (Appendix A). There was no clear difference in the ventilator-associated pneumonia rate (OR 0.33; 95% CI, 0.01–8.83) (Appendix A), but the results were imprecise. None of the included studies reported outcomes, functional status after hospital discharge, or cost associated with the intervention.

Neither of the tests for subgroup effect was significantly different when heterogeneity sources were explored for the outcomes of in-hospital mortality, mechanical ventilation, renal replacement therapy, and SOFA score, according to the definitions of sepsis and septic shock (test for subgroup differences: *p* > 0.05, *I*^2^ < 40%), energy goal (test for subgroup differences: *p* > 0.05; *I*^2^ < 40%), or protein concentration (test for subgroup differences: *p* > 0.05; *I*^2^ < 40%). The planned sensitivity analyses based on the quality of the included studies could not be carried out because all studies were assessed as having unclear or high risk of bias.

## 4. Discussion

Medical nutrition therapy provides the opportunity to compensate for muscle wasting, oxidative stress, and immune response activation, counterbalancing the energy deficit developed during stress and trauma [41,42]; for this reason, it is of great importance for clinicians to know the safety and effectiveness of this intervention in the critical care setting [2,43]. The results of this review support the recommendation issued by some guideline development groups [5,13,44], and the Surviving Sepsis Campaign [2], which advocates the initiation of early enteral nutrition in patients with sepsis or septic shock. Although to this date the certainty of the effect is low, this systematic review documented that EEN may reduce days of mechanical ventilation and SOFA scores in critically ill patients with sepsis or septic shock at the expense of an apparent increase in the frequency of non-serious adverse events (e.g., diarrhea). These findings evoke the known theoretical benefits of this intervention, i.e., modulation of the inflammatory and metabolic response in sepsis, as well as the conclusions of other authors, who argue that early delivery of protein and caloric requirements through the enteral route positively impacts the prognosis of septic patients [6,14].

The results of this review differ from those of a recent publication [45], which did not document a significant benefit with the use of EEN in patients with sepsis. The observed differences between both reviews pertain not only to methodological aspects (e.g., inclusion and exclusion criteria, publication restrictions, number of studies retrieved) but also to the fact that they answer different clinical questions. Even though both reviews share the same target population, Moon et al. included studies with a broader definition of early nutrition (e.g., up to 14 days), implemented the use of late enteral nutrition or early parenteral nutrition as a control, and analyzed other outcomes (e.g., ICU length of stay). These differences explain, at least in part, the substantial heterogeneity observed by Moon et al. and the apparent absence of benefit from EEN [45].

This review has some strengths. Methods were established a priori, and the protocol was registered in advance; strict inclusion and exclusion criteria were implemented, allowing only for the inclusion of studies relevant to the question; the search was broad without publication date or language restrictions; study selection, data extraction, and risk of bias assessment were carried out using validated instruments and in duplicate; the included and excluded studies are described in detail; the quality of evidence was considered at the time of formulating conclusions; and the presence of publication bias was explored.

The low quality of the available evidence is a limitation of this study. The included RCTs have some limitations related to domains such as the randomization process, deviation from the intended intervention, and likelihood of outcome measurement bias. As for the NRSs, although some studies explored the association of interest due to the presence of confounding variables, there is the possibility of residual confusion given the nature of the design. This, added to the limited information regarding the intervention administered, affects our confidence in the observed effect. Finally, another of the limitations of the evidence lies in the presence of manifest imprecision in the width of the confidence intervals. The frequency of events observed for some outcomes (e.g., mortality, ventilator-associated pneumonia) was low, which could mask the presence of a true effect.

Despite these limitations, this review has implications for practice and research. Low-quality evidence suggests that EEN in patients with sepsis and septic shock may reduce days of mechanical ventilation and SOFA scores during follow-up at the expense of an apparent increase in the frequency of non-serious adverse events. Although the frequency of in-hospital mortality was lower in patients with ENN, this difference did not reach statistical significance. Future studies should compare the optimal time for starting this support, analyze the effect of different types of nutritional support (e.g., high-protein-low-calorie versus trophic or enriched diet), and report critical outcomes for the patient and for decision making, such as mortality, quality of life, functional performance, and costs associated with the intervention.

## 5. Conclusions

Low-quality evidence suggested that patients receiving EEN could require fewer days of mechanical ventilation and may show lower SOFA scores during follow-up. EEN could increase the incidence of diarrhea episodes, without an apparent difference in the proportion of patients with ileus or ventilator-associated pneumonia rates. EEN may be a safe and effective intervention for the management of critically ill patients with sepsis or septic shock. Well-designed studies are required to analyze the safety and effectiveness of this intervention.

## Figures and Tables

**Figure 1 nutrients-16-01560-f001:**
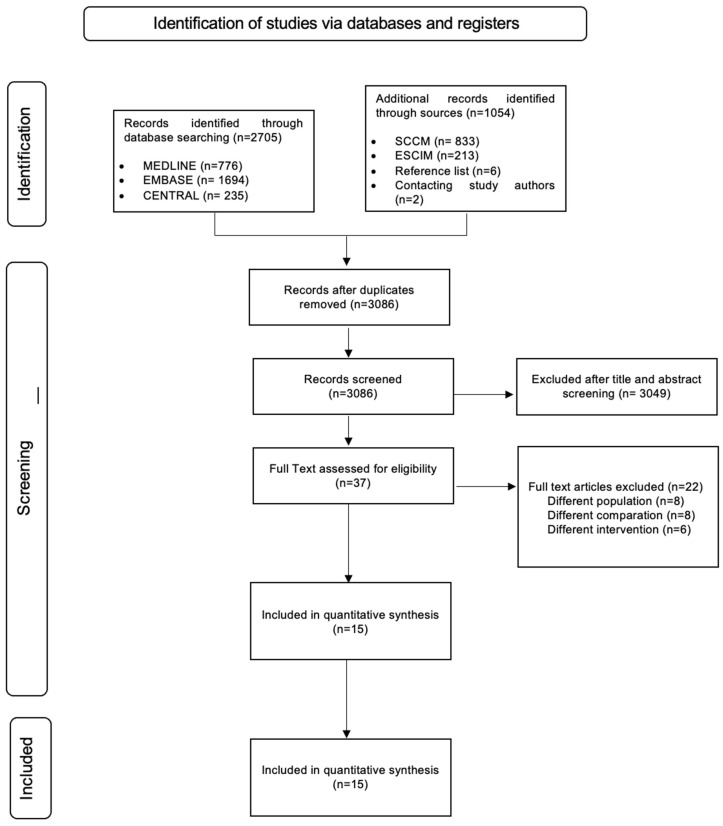
Study flow diagram. Inclusion of studies at different stages for this systematic review and meta-analysis.

**Figure 2 nutrients-16-01560-f002:**
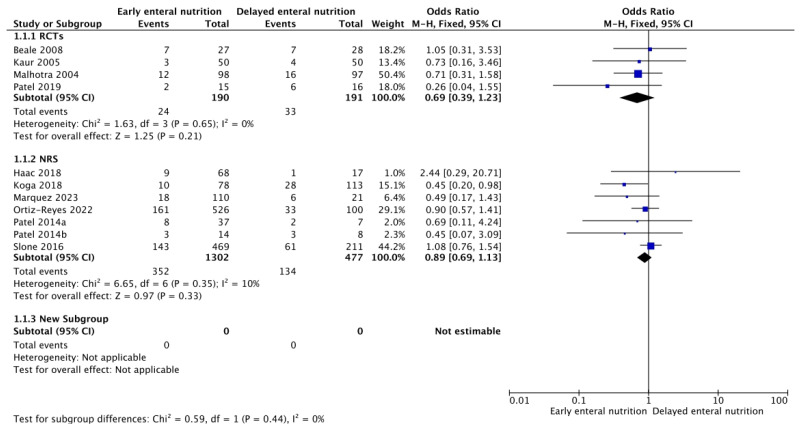
Funnel plot for EEN versus DEN. Outcome: in-hospital mortality [26,27,28,29,31,33,35,36,37,38].

**Figure 3 nutrients-16-01560-f003:**
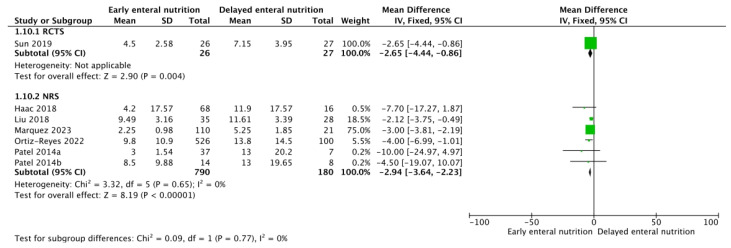
Funnel plot for EEN versus DEN. Outcome: mechanical ventilation [30,31,34,35,36,37].

**Figure 4 nutrients-16-01560-f004:**
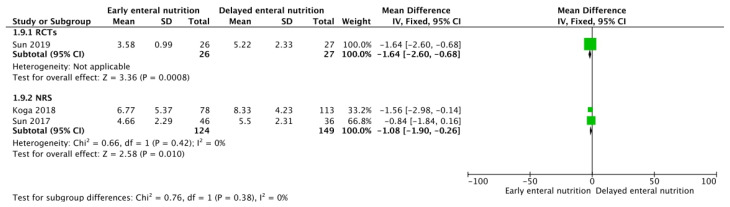
Funnel plot for EEN versus DEN. Outcome: SOFA score [30,33,39].

**Figure 5 nutrients-16-01560-f005:**
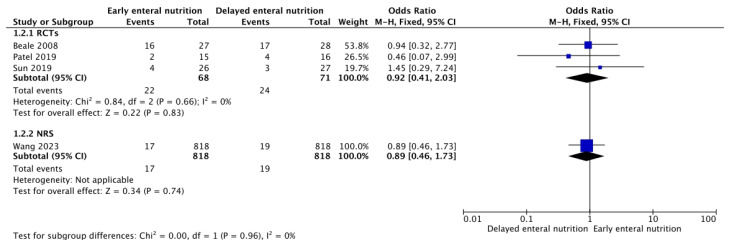
Funnel plot for EEN versus DEN. Outcome: renal replacement therapy [26,29,30,40].

**Figure 6 nutrients-16-01560-f006:**
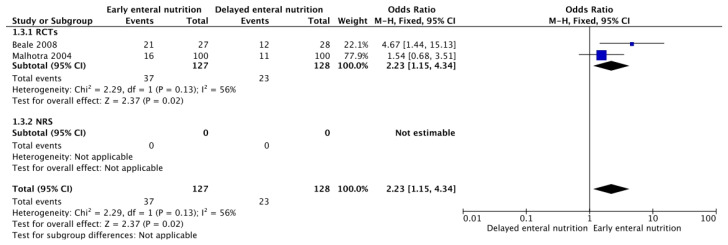
Funnel plot for EEN versus DEN. Outcome: diarrhea [26,28].

## Data Availability

The data that support the findings of this study are available from the corresponding author, [C.F.G.-A. and P.G.-M.], upon reasonable request.

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
