# Peer review of "Early Enteral Nutrition (within 48 h) for Patients with Sepsis or Septic Shock: A Systematic Review and Meta-Analysis"

_nutrients, 2024, doi:10.3390/nu16111560_

Round 1

Reviewer 1 Report

Comments and Suggestions for Authors

Authors study, in a systematic review, the relationship between sepsis or septic shock and early enteral nutrition. Their results indicate a beneficial effect of early enteral nutrition.

The investigation is methodologically well performed.

Strengths and limitations of the study are well indicated.

To improve the manuscript, some minor changes can be considered:

Line 139. Table 3? Please, correct the paragraph.

Line 145. Is a duplicate. Please, delete one.

Line 163. APACHE score is APACHE II? . Please, specify.

Author Response

The investigation is methodologically well performed.

R/. Thanks for your comment. 

Strengths and limitations of the study are well indicated.

R/. Thanks for your comment. 

To improve the manuscript, some minor changes can be considered:

Line 139. Table 3? Please, correct the paragraph.

R/. Thanks. Adjusted.

Line 145. Is a duplicate. Please, delete one.

R/. Thanks. Adjusted.

Line 163. APACHE score is APACHE II? . Please, specify.

R/. Thanks. Adjusted.

Reviewer 2 Report

Comments and Suggestions for Authors

This is well-written review. I have only minor comments.

This review summarized RCT and non-RCT. You need to show this method is validated because it is recommended to do meta-analysis in only RCT.

Your conclusion should have more information. It is very short. 

Your conclusion is early entralnutrition within 48 hours is safe even in septic shock.  You need to discuss why it was safe in septic shock. Maybe some RCT set inclusion criteria and exclusion criteria for safety, for example cathecholamine dose. You need to show how we can manintain the safety. Otherwise, a lot of septic shock patients will experience dange by early eneteral nutriiton.

It is better to clearly show the result of subgroup analysis because the difference between sepsis and septic shock is important, as with protein dose.

Why did you use 48 hours, not 72 hours. Some gourp recommend early enteral nutrition within 72 hours. It is intersting to conduct the analysis within 72 hours.

I  know you added the information in supplemental file, but how about the enteral nutrition timing within 48 hours in each study, for example within 24 hours? 

Author Response

This review summarized RCT and non-RCT. You need to show this method is validated because it is recommended to do meta-analysis in only RCT.

R/. Thanks for your comment. Adjusted. 

Your conclusion should have more information. It is very short. 

R/. Thanks for your comment. Adjusted. 

Your conclusion is early entralnutrition within 48 hours is safe even in septic shock.  You need to discuss why it was safe in septic shock. Maybe some RCT set inclusion criteria and exclusion criteria for safety, for example cathecholamine dose. You need to show how we can manintain the safety. Otherwise, a lot of septic shock patients will experience dange by early eneteral nutriiton.

R/. Thanks for your comment. Adjusted.

It is better to clearly show the result of subgroup analysis because the difference between sepsis and septic shock is important, as with protein dose.

R/. Thanks for your comment. Adjusted.

Why did you use 48 hours, not 72 hours. Some gourp recommend early enteral nutrition within 72 hours. It is intersting to conduct the analysis within 72 hours.

R/. Thank you for your comment. However, we defined early enteral nutrition using 48 hours as a limit in congruence with the recommendation issued by The American Society for Parenteral and Enteral Nutrition (ASPEN), and The European Society for Clinical Nutrition and Metabolism (ESPEN). In this way, given that 48 hours is the usual time to define EEN, we will retain the original proposal.

I  know you added the information in supplemental file, but how about the enteral nutrition timing within 48 hours in each study, for example within 24 hours? 

R/. Thank you for your comment. However, we reviewed the included studies, and all started enteral nutrition in the first 24 to 48 hours without a clear distinction, so it is not possible to make a formal comparison between 24 and 48 hours, which precludes this analysis.